# Desmin and Plectin Recruitment to the Nucleus and Nuclei Orientation Are Lost in Emery-Dreifuss Muscular Dystrophy Myoblasts Subjected to Mechanical Stimulation

**DOI:** 10.3390/cells13020162

**Published:** 2024-01-16

**Authors:** Vittoria Cenni, Camilla Evangelisti, Spartaco Santi, Patrizia Sabatelli, Simona Neri, Marco Cavallo, Giovanna Lattanzi, Elisabetta Mattioli

**Affiliations:** 1CNR Institute of Molecular Genetics “Luigi Luca Cavalli-Sforza”, Unit of Bologna, 40136 Bologna, Italy; vittoria.cenni@cnr.it (V.C.); spartaco.santi@cnr.it (S.S.); patrizia.sabatelli@area.bo.cnr.it (P.S.); 2IRCCS Istituto Ortopedico Rizzoli, 40136 Bologna, Italy; 3Cellular Signalling Laboratory, Department of Biochemical and Neuromotor Sciences, Alma Mater Studiorum, University of Bologna, 40138 Bologna, Italy; camilla.evangelisti@unibo.it; 4Medicine and Rheumatology Unit, IRCCS Istituto Ortopedico Rizzoli, 40136 Bologna, Italy; simona.neri@ior.it; 5Shoulder-Elbow Surgery Unit, IRCCS Istituto Ortopedico Rizzoli, 40136 Bologna, Italy; marco.cavallo@ior.it

**Keywords:** lamin A/C, LINC complex, Emery-Dreifuss Muscular Dystrophy (EDMD), desmin, plectin, mechanical stretching, muscle

## Abstract

In muscle cells subjected to mechanical stimulation, LINC complex and cytoskeletal proteins are basic to preserve cellular architecture and maintain nuclei orientation and positioning. In this context, the role of lamin A/C remains mostly elusive. This study demonstrates that in human myoblasts subjected to mechanical stretching, lamin A/C recruits desmin and plectin to the nuclear periphery, allowing a proper spatial orientation of the nuclei. Interestingly, in Emery-Dreifuss Muscular Dystrophy (EDMD2) myoblasts exposed to mechanical stretching, the recruitment of desmin and plectin to the nucleus and nuclear orientation were impaired, suggesting that a functional lamin A/C is crucial for the response to mechanical strain. While describing a new mechanism of action headed by lamin A/C, these findings show a structural alteration that could be involved in the onset of the muscle defects observed in muscular laminopathies.

## 1. Introduction

Muscle cells respond to mechanical stimuli through a fine-tuned succession of molecular events, which is a basic prerogative for muscle function. When myoblasts are subjected to stretching, the forces propagate within the cells through a dense network of cytoskeletal proteins, including actin, intermediate filaments, microtubules and associated proteins, converging at the nuclear surface. At this level, these forces concentrate at the LINC (Linker of Cytoskeleton and Nucleoskeleton complex), a molecular platform consisting of nesprins, SUN proteins, and emerin, and propagate inside the nucleus reaching lamins, which finally transfer the mechanical information to chromatin [1,2]. A tight relationship between LINC proteins and lamin A/C, and their involvement in nuclear dynamism, has been demonstrated. Lamin A/C is required for the proper localization of nesprin 3 to the nuclear envelope, a mechanism involved in nuclei orientation during fibroblasts migration [3], which also requires a functional lamin A/C-SUN1 complex [4,5]. Moreover, the interaction between lamin A/C and SUN1 is fundamental for nuclear positioning in muscle cells [4,5]. In those cells, lamin A/C is also required for centrosome reorientation and nuclear movement [6]. Interestingly, it has been recently published that A-type lamins modulate the anchoring of LINC complex proteins to the nucleus, influencing cytoskeletal proteins assembly to the nuclear envelope and contractile properties [7]. Upon mechanical strain, lamin A/C contributes to the reorganization of perinuclear actin stress-fibers (known as trans-membrane actin cables or TAN lines) that avoid nuclear deformation [8,9]. Consistent with this evidence, in lamin A/C-deficient cells, including cells from Hutchinson–Gilford Progeria (HGPS) and EDMD2, actin filaments are disorganized [8,10]. Further, *Lmna* -/- fibroblasts show defects in nuclear response to mechanical strain [11].

The importance of the integrity of nuclear lamina and the LINC complex network is highlighted by the occurrence of muscular diseases due to genetic mutations of their components. These disorders include muscular dystrophies caused by mutations of *LMNA* gene, coding for lamin A/C, *EMD* gene coding for emerin, *UNC84 A* and *B* genes coding for SUN1 and SUN2, respectively, and *SYNE1/2* genes coding for nesprin 1 and nesprin 2. Mutations in these genes cause an Emery-Dreifuss Muscular Dystrophy (EDMD) phenotype, characterized by early contractures, cardiomyopathy, and progressive weakness of skeletal muscle [5,12]. Interestingly, mutations of *DES* gene coding for desmin, drive to genetic disorders known as desminopathies, which share a wide range of phenotypical aspects with laminopathies, including musculoskeletal weakness and restrictive cardiomyopathy with arrhythmias or conduction defects [13,14].

Desmin is an intermediate filament specifically expressed in cardiac, skeletal, and smooth muscle cells, where it contributes to muscle mitochondrial homeostasis, sarcomere structure and force transmission [15,16,17]. The existence of a relationship between desmin and lamin A/C has been reported. In murine *Lmna*-deficient cardiomyocytes, the desmin network is disorganized in the perinuclear area and desmin is detached from the nuclear envelope, resulting in a general loss of force transmission during contraction [18,19]. Moreover, desmin expression is down-regulated in *LMNA*-null muscle cells [20], and desmin cytoplasmic aggregates have been observed in *Lmna^H222P^*^/*H222P*^ mice [21]. However, albeit several studies hint at a functional interplay between desmin and lamin A/C, no evidence of a physical interaction between the two proteins has been provided.

In muscle cells subjected to uniaxial cyclic stretch, nuclei are subjected to traction forces that cause their deformation [22] and re-orientation perpendicular to the stretch axis [23]. In addition to proteins of the LINC complex, movement of nuclei in cultured myoblasts is also driven by plectin and desmin, cytoskeletal proteins able to regulate force transmission to the nucleus and nuclear positioning, [24,25]. It has been demonstrated that desmin is recruited to the nuclear envelope of myotubes by plectin, allowing the proper nuclear orientation [26]. Plectin is a giant protein which links and stabilizes actin, microtubules, and intermediate filaments, acting as a modulator of mechanical cells properties [25]. Moreover, plectin 1, the isoform more represented in skeletal muscle cells, is also the main organizer of desmin filaments in myofibers [27]. During mechanical strain, plectin interaction which cytoskeletal proteins is crucial, because by properly distributing tensile forces, it allows nuclear integrity [28]. Interestingly, as observed for desminopathies and laminopathies, diseases due to mutation of plectin gene (*PLEC*), may present a phenotype characterized also by muscular dystrophy [29].

In this work, we investigated the involvement of lamin A/C in cytoskeletal remodelling and nuclear movement of muscle cells during mechanical stress response, focusing on human myoblasts from healthy donors or EDMD2 patients. Our findings show that lamin A/C is able to interact with desmin and plectin, organizing their anchorage to the nuclear surface during mechanical stress response, supporting the role of lamin A/C in the dynamics of human myoblast nuclei subjected to stretching.

## 2. Materials and Methods

### 2.1. Cell Culture, Transfection and siRNA

Human myoblast cultures were obtained from muscle biopsies of skeletal muscle from healthy donors and EDMD2 patients carrying the following *LMNA* mutations: p.Y259D (Patient 1), p.Y259D (Patient 2) and p.L140P (Patient 3). Biopsies were collected in the BioLaM biobank approved by the “IOR Ethics Committee” on 05/09/2016. Prot. gen 0018250-01-13. All EU and local ethical rules were respected. Cell cultures were cultured in Dulbecco’s modified Eagle’s medium (DMEM), supplemented with 20% fetal bovine serum (FBS) (Gibco Life Technology, Thermo Fisher Scientific, Waltham, MA, USA) and antibiotic-anti-mycotic solution (Sigma-Aldrich, St. Louis, MO, USA). Where specified, cells were transfected by Amaxa Nucleofector Technology (Lonza Bioscience, Basilea, Svizzera), with FLAG-tagged plasmids containing: wild-type lamin A, LA-WT, and *LMNA* pathogenetic mutants LA-R527P and LA-R401C, both linked to EDMD2 [30]. Transfection of myoblasts cells was performed using Myrus solution (Promega, Madison, WI, USA)) according to the manufacturer’s instructions and cells were harvested 48 h after transfection. Expression of lamin A/C was silenced with predesigned siRNA: lamin A/C siRNA (SC-3577, Santa Cruz Biotechnology, Dallas, TX, USA) according to the manufacturer’s instructions. Briefly, 50 pmols of siRNA were added to cell cultures for 10 days. Validation of lamin A/C silencing was performed by immunofluorescence analysis.

### 2.2. Mechanical Stretching

The application of mechanical stretching to myoblasts cultures, was performed by a Flexcell^®^ FX- 4000T Tension Apparatus (Flexcell International Corporation, Burlington, NC, USA), a bioreactor applying a mechanical strain to cell monolayers. Uniaxial cyclic tension strain was applied to myoblasts grown on flexible-bottomed culture plates for about 15 days, Collagen I-coated (Dunn Labortechnik GmbH, Asbach, Germany) at 37 °C, 5% CO_2_, with 10% sinusoidal strain, 1 Hz frequency for 4 h.

### 2.3. Antibodies

Antibodies employed were: anti-lamin A/C, goat polyclonal (Byorbit orb37882, Cambridge, UK) used at 1:100 dilution for IF and in situ proximity ligation assay (PLA); anti-lamin A/C (E1, Santa Cruz Biotechnology, Dallas, TX, USA) used at 1:500 dilution for IF and in situ proximity ligation assay (PLA) and 1:2000 for WB; anti-desmin (Abcam Ab15200 Cambridge, UK) used 1:1000 for IF and 1:2000 for PLA and WB; anti-desmin (Chemicon 1:400 for IF; phalloidin (Sigma-Aldrich, St. Louis, MO, USA) 1:1000 for IF; anti H3k9ac (Abcam, Cambridge, UK) 1:200 for IF; anti YAP (Santa Cruz Biotechnology, Dallas, TX, USA) 1:100 for IF; anti-emerin (Monosan, Uden, The Netherlands) 1:100 for IF; anti-FLAG tag (Sigma-Aldrich, St. Louis, MO, USA) 1:1000 for IF; anti-plectin 1 (D6A11, Cell signaling Technology, Danvers, MA, USA) 1:100 for IF and PLA.

### 2.4. Immunoprecipitation

For immunoprecipitation analysis, stretched myoblasts were lysed in high detergent-IP buffer containing: 50 mM Tris-HCl (pH = 7.5), 150 mM NaCl, 0.1% SDS, 1% NP-40, 1 mM DTT, 1.5 mM MgCl, 20 mM NaF, 1 mM PMSF, and protease and phosphatase inhibitors. For each sample, 500 µg of lysate was incubated at 4 °C overnight with 2.5 µg of anti-lamin A/C (E1, Santa Cruz Biotechnology, Dallas, TX, USA) or nonspecific immunoglobulins form Santa Cruz as a negative control. After the addition of 30 μL of protein A/G (Santa Cruz Biotechnology, Dallas, TX, USA) for 60 min at 4 °C, the immunoprecipitated proteins were washed 3 times in IP buffer. Later, the samples were added to Laemmli’s buffer, boiled at 100 °C for 5 min, and subjected to Western blot analysis. Immunoblotted bands were detected by ECL detection system (Thermo Fisher Scientific, Waltham, MA, USA) and an intensity measurement was performed using a Bio-Rad MP Imaging System with Image Lab Touch Software version 3.0.1.14 (Bio-Rad Laboratories, Segrate, Milan, Italy).

### 2.5. In Situ Proximity Ligation Assay

An in situ proximity ligation assay (PLA) was performed using Duolink^®^ In Situ Detection Reagents Orange (DUO92007) (from Sigma-Aldrich) according to manufacturer instructions. Briefly, RT methanol-fixed samples were treated with 4% BSA in PBS to saturate non-specific binding and incubated with primary antibodies overnight at 4 °C. Thereafter, slides were incubated for 1 h at 37 °C with secondary probes diluted to final concentrations of 1:5. Ligation solution was added to each sample and slides were incubated in a humidity chamber for 30 min at 37 °C. Later, ligation solution was removed with wash buffer A and amplification solution was added to each sample. Slides were incubated in a humidity chamber for 100 min at 37 °C and then washed with wash buffers B and for 1 min with 0.01% buffer B. Duolink in situ mounting medium with DAPI was added to the slides and samples were observed by a Nikon Eclipse Ni fluorescence microscope equipped with a digital CCD camera using NIS Elements AR 4.3 software. Quantitative analysis of PLA results was performed using Duolink Image Tool version 1.0.1.2 software (Sigma-Aldrich, St. Louis, MO, USA) by counting 100 nuclei per each of three biological replicates.

### 2.6. Immunofluorescence Analysis

Cells grown on coverslips, and when they reached the confluence, were fixed with absolute methanol at room temperature (about 23 °C) for 10 min or 4% paraformaldehyde 15 min a RT and permeabilized with 0.05% triton for 5 min a RT. After saturation of non-specific binding with 4% BSA solution for 20 min, coverslips were incubated with primary antibodies overnight at 4 °C O.N or 1 h a RT, and revealed with FITC or TRIC-conjugated secondary antibodies diluted 1:200 (incubated for 1 h at RT). Samples were mounted with an anti-fade reagent (Molecular Probes Life Technologies, Monza, Italy) and observed with different microscopes. Quantitative analysis of immunofluorescence, intensity profile and the length of minor and major axes of the nuclei (mm), were performed using NIS-Elements Advanced Research software (Nikon, Minato, Tokyo, Japan), by counting 60 nuclei per each of three biological replicates. Quantitative analysis of immunofluorescence on the nuclear envelope, was performed by selecting the area of the nucleus using NIS-Elements Advanced Research software (Nikon, Minato, Tokyo, Japan).

### 2.7. Imaging and SIM Analysis

Immunofluorescence and PLA analysis were performed using a Nikon Eclipse Ni epifluorescence microscope with a 40×, 60× and 100× objectives. The images captured with NIS- Elements 4.3 AR software were elaborated using Photoshop CS.

Nikon-Structured Illumination Microscopy (3D N-SIM), which permits to observe fluorescent samples at resolutions below the limit the diffraction of light imposes by optical microscopy that is 85–100 nm, was performed using a Plan-Apochromat 100×/1.49 Oil TIRF objective and 488 and 561 nm laser lines as a described in Santi et al. [31]. Image analysis (feature measurements, 3D object count and 3D rendering) was performed using NIS-Elements Advanced Research software (Nikon, Minato, Tokyo, Japan).

### 2.8. Statistical Analysis

Three biological replicates were used in each experiment. Statistical analysis was performed with Student’s *t*-test and statistically significant differences between values are indicated (*p* < 0.05) ± standard deviation ±(SD). For intensity profile analysis we used the Pearson correlation coefficient (r) where r between 0 and 1 is statistically significant.

Nuclei orientation analysis was performed with Fiji-ImageJ plugin OrientationJ (Biomedical Imaging Group, Lausanne, Switzerland). The value of the coherency indicates the degree to which the nuclei are oriented and the value is set from 0.1. to 1.0, where 1.0 is the maximal coherency.

Measurement the radius of the nuclei was performed using NIS-Elements Advanced Research software (Nikon, Minato, Tokyo, Japan).

## 3. Results

### 3.1. Mechanical Stretching Stimulates Desmin and Plectin 1 Translocation to the Nuclear Envelope in Human Myoblasts

During mechanical stress, re-organization of cytoskeletal proteins is necessary for proper remodelling of the protein network that governs nuclear positioning [22]. To investigate the effects of mechanical stress on plectin and desmin localization at the nuclear periphery in human myoblasts, primary cultures from healthy donors were subjected to a uniaxial cyclic mechanical stretching at 1 Hz with 10% of strain. After 4 h, cells were harvested and the localization of desmin and plectin was examined by immunofluorescence analysis.

Under basal condition, in 78% of myoblasts, desmin was localized in the cytoplasm, forming a cage around the nucleus (Figure 1a). However, 23% of myoblasts showed nuclear recruitment of desmin (Appendix A). Upon mechanical stretching, desmin filaments wrapped around the nuclear surface of myoblasts in more than 65% of cells (Figure 1a). It has been demonstrated that plectin acts as a docking site for desmin filaments at the peri-nuclear region [26,28]. Thus we decided to investigate plectin 1 subcellular localization after mechanical stress. In agreement with previous evidence [32], our results show that, under basal conditions, plectin 1 forms a cytoplasmic network in 82% of myoblasts. However, when myoblasts were subjected to mechanical stretching, plectin 1 was observed at the perinuclear region (Figure 1b). Double immunofluorescence staining showed colocalization of plectin 1 and desmin in the cytoskeleton and in portions of the nuclear envelope (Figure 1b).

To investigate if the nuclear recruitment of desmin could be related to myoblast movement, a scratch-wound healing assay was next performed. Confirming this hypothesis, myoblast migration promoted peri-nuclear desmin localization (Appendix A). This result suggests that translocation of desmin is not exclusive of mechanical stress, but it may be also involved in the proper nuclear positioning during myoblast movement.

### 3.2. Mechanical Stress Induces Desmin Recruitment to Lamina A/C in Human Myoblasts

To deepen the molecular mechanisms ruling the nuclear recruitment of desmin, we asked if lamina A/C did interact with desmin upon nuclear recruitment. Immunofluorescence evaluation of lamin A/C and desmin subcellular localizationassessed that while under basal conditions the proteins did not co-localize at the nuclear rim in the vast majority of myoblasts (Figure 2a), mechanical stimulation triggered their co-localization in about 65% of nuclei (Figure 2a). Interestingly, in situ proximity ligation assay (PLA) demonstrated that lamin A/C was able to bind desmin in stressed myoblasts, while a limited number of nuclei were PLA-positive under basal conditions (Figure 2b). In stretched human myoblasts, co-immunoprecipitation experiments confirmed the physical interaction between lamin A/C and desmin (Appendix A). Desmin-lamin A/C interaction in myoblasts subjected to mechanical stress was also confirmed by structured illumination microscopy (SIM) analysis, which showed an increase of desmin-lamin A/C “contact areas” at the nuclear periphery (Figure 2c). Supporting these findings, upon lamin A/C depletion by siRNA, recruitment of desmin to the nuclear envelope was reduced from 65% of nuclei to 30% in stretched myoblasts (Figure 2d and Appendix A).

To understand if lamin A/C was also involved in the nuclear redistribution of plectin 1 observed under mechanical stress, we decided to investigate plectin 1 and lamin A/C subcellular localization after mechanical stress. Immunofluorescence analysis revealed that upon mechanical strain nuclear plectin1 co-localized with lamin A/C (Figure 2e). Interestingly, plectin 1- lamin A/C interaction was also demonstrated by PLA (Figure 2f). As observed for desmin, plectin 1- lamin A/C PLA signals were mostly detected after mechanical stimulation (Figure 2f), while a low signal per nucleus, was detected under basal conditions with an overall increase of 64% relative to non-stimulated cells (Figure 2f). These findings demonstrated the involvement of lamin A/C, in the reorganization of cytoskeletal protein scaffold at the nuclear envelope during mechanical stretching response.

### 3.3. The Recruitment of Desmin to the Nuclear Envelope Is Altered in EDMD2 Myoblasts

Considering the dynamic nature of the interaction between lamin A/C and desmin, we investigated the possibility that EDMD2-causing *LMNA* mutations could affect desmin recruitment to the nuclear envelope. For this purpose, myoblasts from healthy donors and from 3 different EDMD2 patients carrying the Y259D (patient 1 and 2) or L140P (patient 3) mutations of lamin A/C, were subjected to mechanical strain or left unstimulated, and then precessed for immunofluorescence analysis for desmin detection. In unstimulated EDMD2 samples, the number of myoblasts showing desmin recruitment to the nuclear envelope was reduced compared to healthy controls (Figure 3a). Similarly, when myoblasts were subjected to mechanical stretching, the percentage of EDMD2 cells with desmin recruitment to the nuclear rim was significantly lower compared to controls (15%,16% and 19% for each EDMD2 cell line out of 55% for controls) (Figure 3b). Moreover, 35% of EDMD2 myoblasts showed desmin disorganization also in the cytoplasm, both under basal conditions and after cyclic stress (Appendix A). Cytoskeleton disorganization also involved actin stress fibers, as demonstrated by altered phalloidin staining observed in 23% of unstimulated EDMD2 myoblasts and 37% of EDMD2 muscle cells subjected to cyclic stretching (Appendix A).

To evaluate the above reported interplay in myoblasts carrying other disease-causing *LMNA* mutations, human control myoblasts were transfected with FLAG-Lamin A R527P and FLAG-Lamin A R401C or FLAG-Lamin A wild-type. Immunofluorescence analysis of FLAG-tagged proteins and desmin, allowed us to evaluate the subcellular localization of overexpressed lamin A and endogenous desmin in myoblasts subjected to mechanical stretching. Results shown in Figure 3c revealed that desmin was localized at the nuclear envelope in 40% of stretched myoblasts expressing wild-type lamin A, whereas it was not recruited to the nuclear periphery in myoblasts expressing *LMNA*-mutants (Figure 3c). These results showed that non-functional lamin A/C affected desmin nuclear recruitment upon mechanical stretching independently of the *LMNA* mutation.

To address the role of the LINC complex in the lamin A/C interplay with desmin and plectin 1, we analysed SUN1, nesprin1, emerin and lamin A/C levels and interaction. Immunofluorescence analysis revealed that protein amount did not change in EDMD2 myoblasts compared to controls, neither under static conditions or upon stretching (Figure 3d, Appendix A). However, a reduction of the binding between lamin A/C and SUN1 was measured by PLA in laminopathic myoblasts both in basal conditions and upon mechanical stretching (Figure 3d).

### 3.4. LMNA-Mutated Myoblasts Subjected to Mechanical Stretching Show Reduced Recruitment of Plectin 1 to the Nuclear Envelope and Nuclei Reorientation Defects

Interestingly, immunofluorescence analysis of myoblasts from control and EDMD2 myoblasts subjected to mechanical stress revealed that lamin A mutations also impaired plectin 1 recruitment to the nuclear periphery (Figure 4a).

Reported data have demonstrated that after mechanical stretching nuclei are oriented perpendicularly to the strain direction [23], as shown in Figure 4b. Quantification of nuclei orientation was reported as a measure of coherency in the graph, where coherency indicates the degree to which the nuclei are oriented. Interestingly, in control myoblasts subjected to cyclic stretching, we also observed a reduction of H3K9 acetylation, suggesting heterochromatin remodelling (Appendix A).

To evaluate if mutations of lamin A/C might impair nuclei orientation, control and EDMD2 myoblasts were exposed to uniaxial stretching. In stretched EDMD2 myoblasts carrying lamin A/C mutations, the majority of the nuclei (about 60%) lost proper anisotropic rearrangement, acquiring an orientation parallel to stretch direction (Figure 4c). Since it has been published that YAP-mediated mechanosensitive response is altered in laminopathic myoblasts [9], we performed YAP immunolocalization in EDMD2 myoblasts and healthy control before and after cyclic stretching. In EDMD2 myoblasts under basal conditions, the nuclear fluorescence intensity of YAP was increased with respect to healthy controls (Figure 4d). Furthermore, we observed YAP translocation to the nucleus following cyclic stress in control but not in laminopathic myoblasts (Figure 4d).

We also found morphological alterations of the nucleus in EDMD2 myoblasts. The major axis of the nucleus appeared to be increased under basal conditions in EDMD2 cells relative to controls (Figure 4e). Upon stretching, in both control and EDMD2 myoblasts, we observed an increase of the main diameter of the nuclei with respect to the minor diameter, compared to unstretched cells (Figure 4e). However, much more significant enlargement of the major axis of nuclei was observed in stretched laminopathic myoblasts than in healthy controls (Figure 4e).

## 4. Discussion

Several studies demonstrated an involvement of lamin A/C in mechanosignaling [1]. It has been shown that lamin A/C contributes to response to tensile forces, reinforcing the nucleus-cytoskeleton scaffold organization, through the LINC platform, but the mechanism underlying this molecular rearrangement in muscle cells remains still elusive [1,7,33].

In this study, we observed that, in myoblasts, an intact lamin A/C is necessary for desmin recruitment to the nuclear envelope during the response to mechanical strain. In fact, both EDMD2 myoblasts carrying *LMNA* mutations, as well as human myoblasts transfected with EDMD2-causative lamin A/C mutants, showed defective nuclear recruitment of desmin, and similar results were obtained in stretched-myoblasts upon *LMNA* silencing. Consistent with this observation, it has been demonstrated that cardiomyocytes lacking lamin A/C show desmin detachment from the nuclear surface [19], while lamin A-deficient mouse embryonic fibroblasts fail to properly organize vimentin, another intermediate filament, around nucleus [7].

Since desmin has been involved in nuclear dynamics through its binding with plectin [26], we also investigated plectin 1, the isoform mainly expressed in muscle cell. Our findings revealed that, in myoblasts expressing pathogenetic lamin A mutants, plectin 1 is not correctly recruited to the nuclear rim upon mechanical stimulation. Moreover, we observed that while in control stretched myoblasts, nuclei were oriented perpendicularly to the strain direction [22,23]. In stretched laminopathic myoblasts, nuclei failed to correctly align. Altogether, these results support the hypothesis that lamin A/C modulate desmin-plectin complex at the nuclear level, to regulate the spatial orientation of nuclei in response to mechanical stimulation, and confirm the centrality of the role of lamin A/C in coordinating repositioning of nuclei [33,34].

It has been demonstrated that deficiency of plectin affects heterochromatin organization thus influencing nuclear stiffness [26]; on the other hand, lamin A/C involvement in epigenetic modifications has been widely reported in myoblasts [4,31]. Our results here presented allow us to speculate that lamin A/C-plectin interaction also influences chromatin dynamics and nuclear stiffness during mechanical stress response. Supporting the relationship between cytoskeletal and chromatin changes, we observed a reduction of H3K9 acetylation in stretched myoblasts.

Of note, we also observed actin disorganization in the perinuclear cap, as previously reported [8]. This suggests that the whole cytoskeletal interplay with the nucleoskeleton is affected by *LMNA* mutations in stretched myoblasts, possibly also through feedback mechanisms. In this respect, it is worth noting that the mechanotransducer YAP was unable to relocalize in nuclei upon cyclic stretching, as previously shown [9].

## 5. Conclusions

Cytoskeletal organization at the nuclear level is driven by the molecular axis made up of desmin, plectin, nesprin 3 and SUN1 [35], while SUN1 depletion affects the reorganization of cytoskeletal proteins as actin and vinculin [35]. In EDMD2 fibroblasts, SUN1 mutations increase the severity of nuclear defects and microtubule association with the nuclear envelope [5]. Most importantly, in myotubes, SUN1 deficiency, either caused by *LMNA* or *SUN1* mutations [4,5], induces myonuclear clustering in part due to impaired microtubule dynamics [4,5]. We cannot rule out the possibility that the altered SUN1 interplay with lamin A/C, here observed in EDMD2-myoblasts, both under basal conditions and after mechanical stretching, could cause reduced SUN1 recruitment to myotube nuclei [4]. Notwithstanding, based on our results, lamin A/C-SUN1-desmin dynamic interactions appear to be major determinants of muscle mechanosignaling. In support of a major cross-talk between lamin A/C and desmin in this context, is the reported increase of nucleoplasmic lamin A/C in desmin-null cells [13].

However, data reported in our study indicate that a functional nuclear lamina is necessary to properly arrange cytoplasmic intermediate filaments in response to mechanical stretching, while alterations of this mechanism occurring in *LMNA*-mutated myoblasts impair nuclear positioning already in mononucleated muscle precursors and could be crucial to the onset of skeletal muscle defects observed in laminopathies.

## Figures and Tables

**Figure 1 cells-13-00162-f001:**
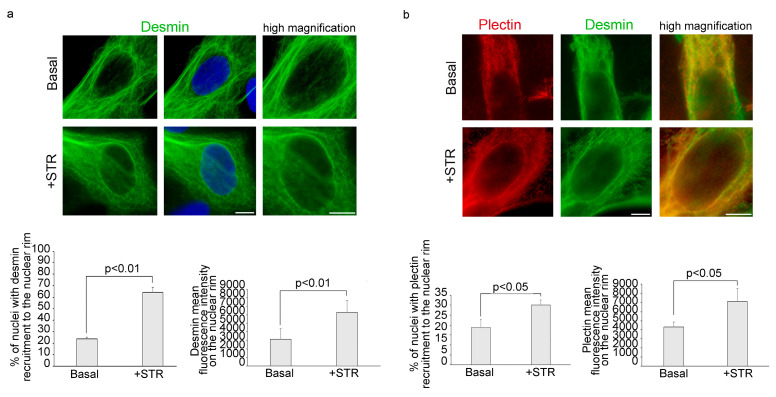
Desmin and plectin 1 recruitment to the nuclear rim increases after cyclic mechanical stretching. Immunofluorescence analysis of cytoskeletal proteins was performed in human myoblasts under basal conditions (Basal) or upon cyclic stress (+STR). (**a**) Immunostaining and high magnification of desmin (green). 4,6-diamidino-2-phenylindole (DAPI, blue) was used to counterstain cell nuclei. The percentage of cells showing desmin recruitment to the nuclear rim and desmin mean fluorescence intensity on the nuclear envelope was reported in the graphs below. (**b**) Double immunofluorescence analysis of plectin 1 (red) and desmin (green) and high magnification of merge image on the right. Statistical analysis of plectin recruitment to the nucleus and plectin mean fluorescence intensity on the nuclear envelope was reported in the graphs below. Scale bars, 10 μm. Three biological replicates were used in each experiment and statistically significant differences (*p* < 0.05 or *p* < 0.01) between values are indicated.

**Figure 2 cells-13-00162-f002:**
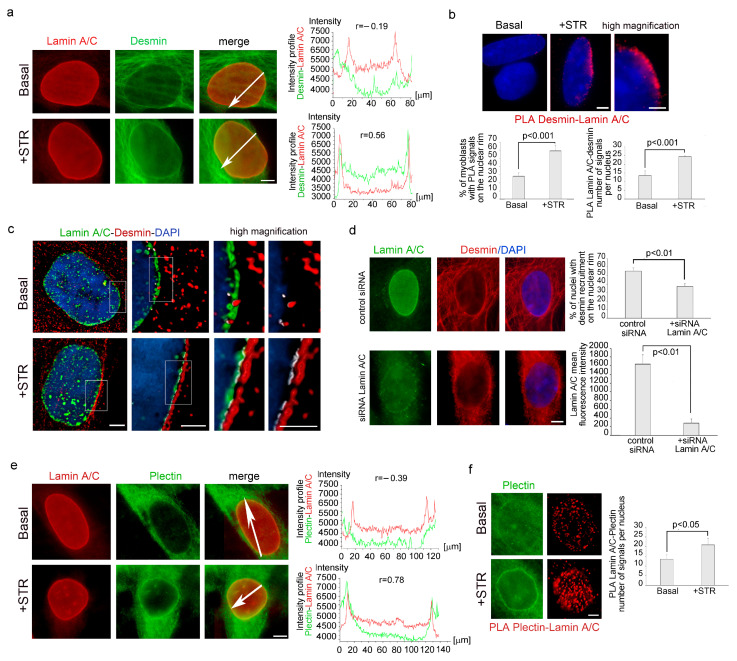
Lamin A/C binds both desmin and plectin 1 in myoblasts subjected to mechanical stress. (**a**) Immunofluorescence analysis of lamin A/C (red) and desmin (green) in human myoblasts unstretched (Basal) or subjected to mechanical stretching (+STR). Intensity profile of the line under the arrows was reported on the right. (**b**) PLA of desmin and lamin A/C (red dots) in myoblasts under basal (Basal) or stretched conditions (+STR). A graphical presentation of the quantitative analysis of PLA signals was reported in the graphs below. (**c**) Structured illumination microscopy of lamin A/C (green) and desmin (red) in myoblasts left unstretched or exposed to mechanical stretching. White signals are generated by colocalization of lamin A/C and desmin. (**d**) Immunofluorescence analysis of desmin (red) and lamin A/C (green) in control siRNA and after silencing of lamin A/C (siRNA Lamin A/C) of human myoblasts subjected to mechanical stretching. Statistical analysis of the percentage of nuclei with desmin recruitment to the nuclear rim was reported in the low graph. (**e**) Immunofluorescence analysis of lamin A/C (red) and plectin 1 (green) in human myoblasts unstretched (Basal) or subjected to mechanical stretching (+STR). Intensity profile of the line under the arrows was reported on the right. (**f**) PLA of plectin 1 and lamin A/C (red dots) and plectin 1 immunofliorescence in human myoblasts unstretched (Basal) or subjected to mechanical stretching (+STR). Graphical analysis of PLA signal was reported in the graph on the right. DAPI (blue staining) was used to counterstain cell nuclei in the figures (**b**–**d**). Scale bars, 2 μm. Three biological replicates were used in each experiment and statistically significant differences (*p* < 0.01 or *p* < 0.001) between values are indicated. The Pearson correlation coefficient (r) was used in figure (**a**,**e**) (r between 0 and 1 is statistically significant).

**Figure 3 cells-13-00162-f003:**
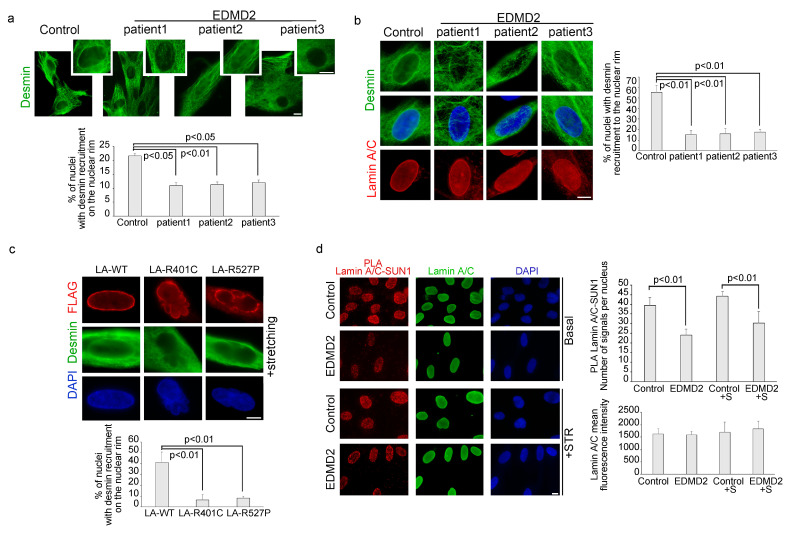
Mutated lamin A/C is unable to recruit desmin during mechanical stretching. (**a**) Immunofluorescence analysis of desmin in healthy donor and EDMD2-myoblasts in basal conditions. High magnification on the top right. Statistical analysis of % of nuclei with desmin recruitment to the rim was reported in the graph below. (**b**) Immunofluorescence analysis of desmin (green) and lamin A/C (red) in stretched myoblasts from control and EDMD2 myoblasts. % of nuclei with desmin recruitment to the rim was reported in the graph on the right. (**c**) Human myoblasts transfected with FLAG LA-WT, FLAG LA-R401C, FLAG LA-R527P were stained with anti-FLAG antibody (red) and desmin (green). Percentage of transfected cells with desmin nuclear localization at the nuclear envelope was reported in the graph below. (**d**) PLA of SUN1 and lamin A/C (red dots) and lamin A/C immunofluorescence (green) in human myoblasts unstretched (Basal) or subjected to mechanical stretching (+STR) from control and EDMD2 patient. A graphical presentation of the quantitative analysis of PLA signals and lamin A/C mean fluorescence intensity were reported in the graphs on the right. DAPI (blue staining) was used to counterstain cell nuclei in figure (**b**–**d**). Scale bars, 10 μm. Three biological replicates were used in each experiment and statistically significant differences (*p* < 0.05 or *p* < 0.01) between values are indicated.

**Figure 4 cells-13-00162-f004:**
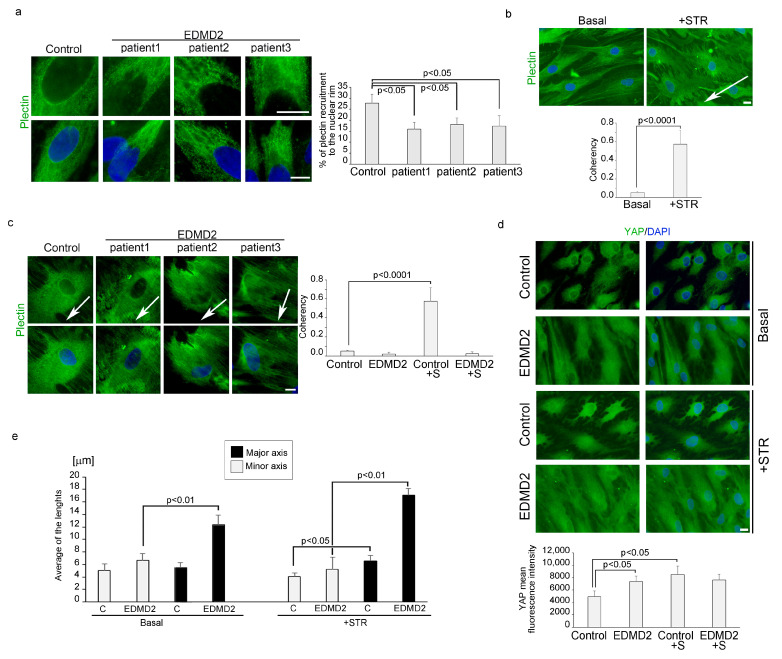
Defective plectin 1 recruitment to the nuclear periphery and altered nuclei reorientation occur in EDMD2 myoblasts after mechanical stretching. (**a**) Immunofluorescence analysis of plectin 1 (green) in myoblasts from healthy donor (control) and EDMD2 myoblasts (EDMD2). Percentage of nuclei with plectin 1 recruitment to the rim was reported in the graph on the right. (**b**) Human myoblasts from control in basal condition (basal) or after mechanical stretching (+STR) subjected to immunofluorescence of plectin 1 (green). Quantification of nuclei orientation was reported in the graph below. (**c**) Immunofluorescence analysis of plectin 1 (green) in myoblasts from healthy donor (control) and EDMD2 myoblasts (EDMD2) subjected to mechanical stretching. In the graph on the right, was reported the nuclei orientation analysis. White arrows in figure b and c indicate the strain direction. (**d**) Immunofluorescence analysis of YAP (green) in myoblasts from healthy donor (control) and EDMD2 myoblasts (EDMD2). Average of YAP fluorescence intensity was reported in the graph below. (**e**) Quantitative measurement of the major axis of the nuclei respect to the minor axis, in control (C) and EDMD2 myoblasts (EDMD2) before (basal) and after cyclic stretching (+STR). DAPI (blue staining) was used to counterstain cell nuclei figures (**a**–**d**). Scale bars, 10 μm. The coherency value is set from 0.1. to 1.0, where 1.0 is the maximal coherency. (*p* < 0.0001). Three biological replicates were used in each experiment and statistically significant differences (*p* < 0.05 or *p* < 0.01) between values are indicated.

## Data Availability

Data supporting reported results can be provided by the authors upon reasonable request.

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
