# Peer review of "Desmin and Plectin Recruitment to the Nucleus and Nuclei Orientation Are Lost in Emery-Dreifuss Muscular Dystrophy Myoblasts Subjected to Mechanical Stimulation"

_cells, 2024, doi:10.3390/cells13020162_

Round 1

Reviewer 1 Report

Comments and Suggestions for Authors

In the current study, the authors examined how the absence of lamin A/C from the LINC complex results in aberrant localisation of the intermediate filaments, plectin and desmin via having utilized human myoblasts sourced from either healthy or EDMD patients, harbouring LMNA mutations, p.Y259D and p.L140P. Primarily, the authors characterized the recruitment of either desmin or plectin to the nuclear rim via immunofluorescence and reported this as percentage recruitment. The authors demonstrated that mutations to the LINC complex disrupted this recruitment. Moreover, they demonstrated that recruitment is further disrupted when subjected to mechanical stretch. However, the authors have not provided sufficient data to show the consequences of myoblasts that unable to localize desmin/plectin and whether this is the basis for disease phenotype. There are a number of concerns that need to be addressed and clarified to support their conclusions, as outlined below:

Major points

1.      The study relies primarily on the percentage recruitment of desmin/plectin. Based on this analysis alone the claims are too strong. Strongly suggest characterizing the nuclear morphology. Particularly as the authors write at Line 50-51 “Lamin A/C activity is also crucial for protecting the nuclear architecture following mechanical stretching”,  yet have provided no evidence supporting this conclusion at current stage.

2.       Performing immunofluorescence intensity analysis to further support percentage recruitment to nuclear rim. Will need to demonstrate consistent microscopy parameters if doing immunofluorescence intensity analysis.

3.      Need to define what the radius of the nuclear rim is here. Was this maintained throughout the assays and how was this determined?

4.      Need to demonstrate the absence of lamin A/C within the patient samples and the converse for the control samples. To include immunofluorescence and western blots. Furthermore, consider demonstrating the localisations of other central LINC complex proteins including SUN1/2, nesprin 1, and emerin.

5.      Need to demonstrate siRNA knockdown efficiency via western blots. To include scrambled control.

6.      Figure 2B and Figure 2E: Correlation needs to be quantified. Consider including Pearson’s correlation coefficient here. Also, remove DAPI from the intensity profile plots.

7.      No more details for Statistical analysis in either the methods section or the figure legends. No efficient to write “Statistical analysis was reported” without detailing what was performed. Could include the following: number of nuclei analysed, number of experimental repeats, which statistical analyses were performed, post-hoc testing, was normality testing performed, whether mean ± SEM or mean ± SD are reported.

8.      Majority of figures are overexposed and not in quality. For example, Figure 1A “+STR Merge” the fibrillar detail of desmin is lost due to poor imaging. Consider replacing with good resolution figures.

9.      To further support mechanical stretching experiments, consider examining mechanosensitive proteins including YAP localisation. This may provide a better understanding for the mechanisms at play.

Minor points:

1.      Line 45: “Moreover, the relation of lamin A/C…”, suggest replacing with “relationship”.

2.      Line 62: “Mutations in these genes induce” Consider replacing with “cause”.

3.      Line 63 and Line 67: “Cardiomyopathy”. Could you specify which sub-type here.

4.      Line 64-67: Consider moving this section to that after Line 73 “…have been observed in LmnaH222P/H222P mice [17]”.

5.      Line 81: “nuclear dynamism” . Please briefly define what is meant here.

6.      Line 86: “represented in muscle cells”.  Is this the case for all muscle sub-types? If in reference to skeletal and cardiac muscle, please state striated muscle.

7.      Line 87-88: “Plectin is a giant…cells properties” Consider moving this to the beginning of (Line 85) “Moreover, plectin 1, the isoform” to make the text clearer.

8.      Line 90 and Line 192: “Tension forces”, consider replacing with tensile forces, and also discussing this point further.

9.      Line 122: “Myoblasts grown on flexible-bottomed culture plates”.  For how long were myoblasts cultured?

10.  Replace commas with full-stops for decimals. For example, Line 135: “1,5mM MgCl” should read 1.5mM MgCl. Replace for all instances.

11.  Line 140: “Boiled”.  At what temperature and for how long?

12.  Line 161: “hot absolute methanol”.  Specify temperature.

13.  Line 181-182: “During mechanical stress response, cytoskeletal proteins re-organization is necessary for a proper nuclear network remodelling…” Replace with the following:

“During mechanical stress, cytoskeletal protein re-organization is necessary for proper nuclear network remodelling…”

14.  Line 186: “explored”.  Consider replacing with “examined” or “determined”.

15.  Line 187: “Majority of the myoblasts”. Need to state the percentage of myoblasts here (as you have done in Line 188).

16.  Line 188: “Forming a basket”. Would be better to replace this with “cage”, as a basket implies an open top whereas cage suggests an encased object.

17.  Line 190: “Lean on”. Is this correct phrasing? Please clarify this statement.

18.  Line 193: “able to reorganize desmin filaments”. Could you please describe how.

19.  Line 195: “Our results showed that in the most of myoblasts”. Need to state percentage here. Also, remove “the”.

20.  For figures: consider removing gridlines from graphs.

21.  Line 204: “mechanical stress”.  Consider replacing with “cyclic stretch”.

22.  Figure 1b: Plectin appears to approach the nucleus unilaterally. Is this the case or is the imaging poor? Please replace with more representative figure.

23.  Line 256-259: “As observed for desmin…detected under basal conditions”. Would insert percentage changes here.

24.  Line 271: “The percentage of EDMD2 cells”. Specify what percentage here.

25.  Figure 3: Nuclear morphology looks particularly changed in response to stretching. Consider quantifying nuclear morphology changes. Percentage recruitment alone is not sufficient.

26.  Line 309-310: “Desmin nuclear recruitment after mechanical stretching was reduced also in myoblasts”. This is unclear and unsupported by the data. Need to quantify immunofluorescence intensity or carry out fractionated western blots to determine the quantity of protein present. Given that the majority of figures are overexposed also makes interpretation difficult.

27.  Line 310, please add reference for this EMD mutation.

28.  Figure 4: Would introduce a muscle biomarker such as actinin or phalloidin for actin to demonstrate the architecture of the muscle.

29.  Figure 4: Orientation can be quantified using Coherency within ImageJ Fiji. Consider quantifying the orientation here.

30.  Line 342: “anomalous”. Do you mean to say “anisotropic” here?

31.  Line 342: “Parallel”. Do you mean “perpendicular”?

32.  Line 366: “Nuclei failed to correctly align”. This was not quantified in the results. Remedy the lack of quantification here to support this statement.

33.  Figure S2, consider showing the Whitefield picture for would healing as well. 

Comments on the Quality of English Language

   Line 80: “reorentation” , please check spelling.

Reviewer 2 Report

Comments and Suggestions for Authors

The authors aimed to examine the functional role of desmin, lamin, and plectin proteins in nuclei reorientation upon applying tension in normal and muscular dystrophy myoblasts.

The experiments were well conducted.  However, I raised only a few comments for general readers.

1) How did the authors stretch myoblast in culture, continuous or repetitive etc? Please describe in detail in Methods section.

2) Numbers of nuclei with specific protein recruitment are shown in column graph. The readers want to see the field images how the several nuclei with or without specific protein recruitment appear under stretch or basal conditions. Please show them in the Supplementary Materials.

3) How is the effect of reorientation of nuclei under stretch, breaking resistant or transcription change, or DNA damage ?

4) The authors should show the cartoon how the desmin, lamin, and plectin proteins work for nuclear reorientation under mechanical stretch and fails in the myoblast of mutants or dystrophy patients. Many cytoskeletal proteins as well as these three proteins may maintain the nuclear orientation. What is the most important and works as a master switch for nuclear reorientation? Please comment in the text.

Round 2

Reviewer 1 Report

Comments and Suggestions for Authors

The authors answered all the important questions, and the manuscript has been significantly improved.  

There is a minor point, which would good to address if possible: 

The Figure 1B (+STR, Plectin) was stitched together, would it be good to add one of them as an insert to show it? 

Author Response

There is a minor point, which would good to address if possible: 

The Figure 1B (+STR, Plectin) was stitched together, would it be good to add one of them as an insert to show it? 

We appreciate the suggestion of the Reviewer and we replaced the picture in the figure 1B (+STR plectin), with a more representative image to better show plectin arrangement around the nucleus.